# Detection of Porcine Circovirus Type 3 in Free-Ranging Wild Boars and Ticks in Jiangsu Province, China

**DOI:** 10.3390/v17081049

**Published:** 2025-07-28

**Authors:** Fanqi Sun, Meng Li, Yi Wang, Wangkun Cheng, Meirong Li, Changlin Deng, Xianwei Wang, Zhen Yang

**Affiliations:** 1Key Laboratory of Animal Diseases Diagnostic and Immunology, Ministry of Agriculture, MOE International Joint Collaborative Research Laboratory for Animal Health & Food Safety, The Belt and Road International Sci-Tech Innovation Institute of Transboundary Animal Disease Diagnosis and Immunization, College of Veterinary Medicine, Nanjing Agricultural University, Nanjing 210095, China; sunfanqi@stu.njau.edu.cn (F.S.); limeng@njau.edu.cn (M.L.); 2020107081@stu.njau.edu.cn (Y.W.); xwwang@njau.edu.cn (X.W.); 2Nanjing Hongshan Forest Zoo, Nanjing 210028, China; 20070590cwk@163.com (W.C.); mr1971@126.com (M.L.); dchanglin75@126.com (C.D.)

**Keywords:** PCV3, wild boar, detection, genome sequence analysis, ticks

## Abstract

Porcine circovirus type 3 (PCV3) has been detected in wild boars across many countries in Europe, Asia, and South America. However, data regarding the presence of porcine circoviruses in wild boars and ticks remain limited. In this study, we investigated the presence and genetic characteristics of PCV3 in wild boars and parasitizing ticks in Jiangsu, China. Samples, including whole blood, serum, tissues, feces, and oral fluids from wild boars, as well as ticks collected from 47 wild boars, were obtained between March 2021 and November 2022. PCR results indicated that 34.0% (16/47) of wild boars tested positive for PCV3, while ELISA detected 41.9% (18/43) seropositivity. RT-qPCR results showed that 7.2% (6/83) were positive for PCV3 in 83 analyzed tick samples, with all positive samples identified as *Amblyomma testudinarium*. The PCV3 genome obtained from wild boars was classified as PCV3a and was closely related to the strain identified in domestic pigs in Nanjing, Jiangsu Province. Collectively, these findings confirm the presence of PCV3 in wild boars in Jiangsu and suggest a possible link of PCV3 infection among domestic pigs, wild boars, and ticks, providing new insights into the transmission risk of PCV3 at wildlife–livestock–human interfaces and highlighting the genetic homology between strains from wild and domestic pigs.

## 1. Introduction

Porcine circovirus (PCV) belongs to the genus *Circovirus* within the family *Circoviridae* and is recognized as the smallest virus known to infect mammals. To date, four circoviruses have been clinically identified in pigs: PCV1, PCV2, PCV3, and PCV4. PCV3 was first detected in commercial swine farms in the United States in 2015 [1,2] and has since been reported worldwide [3,4]. Although many PCV3 infections appear to be subclinical [5,6,7], clinical cases have been associated with reproductive failure [8,9,10], respiratory signs [11,12,13], gastrointestinal disorders [13,14], porcine dermatitis and nephropathy syndrome (PDNS) [15], and multi-systemic inflammation [2,16]. Several studies have attempted to investigate the causal relationship between PCV3 infection and clinical signs by inoculating the virus into specific pathogen-free pigs or cesarean-derived, colostrum-deprived piglets [17,18,19]. However, inconsistent findings regarding clinical signs and histologic lesions were reported following experimental PCV3 infection [20].

Wild boar can serve as reservoirs for several important swine diseases, such as African swine fever, classical swine fever, porcine reproductive and respiratory syndrome, foot-and-mouth disease, and pseudorabies [21,22,23]. In recent years, global population growth, intensified agricultural practices, suburban expansion, and global deforestation have increased interactions between wild boars and domestic pigs, thereby raising the risk of pathogen transmission between the two [21]. PCV3 has also been detected in wild boars in Italy [5,24,25], Spain [26], Germany [27], Brazil [28,29], Japan [30], Austria [31], and South Korea [32]. In China, a recent study identified PCV3 in wild boars collected in 19 regions of China during ASF surveillance, reporting a positivity rate of 10.9% (27/247) [33]. Nevertheless, few studies characterized the epidemiological features of PCV3 in wild boars in urban landscapes, where complex wildlife–livestock–human interfaces exist.

Although PCV3 has not been considered a tick-borne disease, emerging evidence suggests that ticks may play a role in its transmission. Ticks, as obligate blood-feeding ectoparasites, rank second only to mosquitoes among arthropod vectors that transmit various pathogens to vertebrates worldwide [34]. A previous study detected PCV3 in *Ixodes ricinus* ticks collected from PCV3-negative roe deer [35]. However, the role of tick infestation in wild boar and its potential contribution to the epidemiology of PCV3 remain poorly understood.

Nanjing, the capital of Jiangsu Province, is located in eastern China and covers 11 districts with a total area of 6587.02 square kilometers [36]. The city hosts more than 10 national and provincial parks that provide suitable habitats for wild boars [37]. Reports of human–wild boar conflicts have increased in recent years, highlighting the growing presence of wild boars in urban and peri-urban areas [38]. In addition, there are 70 pig farms with more than 50 pigs each in Nanjing (data provided by the local veterinary authority), further facilitating potential contact between wild boars and domestic pigs.

This study aimed to investigate the presence and genetic characteristics of PCV3 in wild boars and ticks parasitizing wild boars in Nanjing, Jiangsu Province, China.

## 2. Materials and Methods

### 2.1. Sample Collection

Nanjing Hongshan Forest Zoo Wildlife Rescue Station is authorized by the government to capture wild boars in Nanjing. When wild boars are sighted in high-traffic areas or residential neighborhoods where public safety may be at risk, veterinarians from the Rescue Station are called upon to assist local police by anesthetizing the animals and transferring them to the station isolation unit, where they are quarantined for at least three weeks. After the quarantine period, surviving wild boars are released into nearby forests, away from human activity. Some wild boars were found dead due to traffic accidents or drowning, while others died in the isolation unit as a result of previous trauma.

From March 2021 to November 2022, blood samples were collected from wild boars captured in urban areas of Nanjing by veterinarians from Nanjing Hongshan Forest Zoo after dart-induced anesthesia. Whenever possible, ticks attached to the animals or found on their skin surface, as well as fecal samples and oral fluids, were also collected. In cases where wild boars died, tissue samples including heart, liver, spleen, lung, and kidney were collected from the carcasses. All samples were submitted to the Veterinary Diagnostic Laboratory of Nanjing Agricultural University for PCV3 testing. Information, including sex, date of capture, and the precise latitude and longitude of the capturing site, was recorded. Capturing locations were mapped using QGIS 3.20.2 (Open Source Geospatial Foundation Project, http://qgis.osgeo.org, accessed on 15 February 2025).

### 2.2. Sample Preparation

Blood samples were centrifuged at 3000× *g* for 15 min, and the resulting serum was stored at −80 °C until further processing. Tissue homogenates, prepared by pooling heart, liver, spleen, lung, and kidney samples, as well as fecal samples, were diluted tenfold with phosphate-buffered saline (PBS; 0.1 M, pH 7.4), mixed with grinding beads, and homogenized using a Rapid Grinder apparatus (Jingxin, Shanghai, China) at 60 Hz for four times of 90 s each. Following three freeze–thaw cycles, the tissue homogenates were centrifuged at 6000× *g* for 10 min, and the supernatants were collected for DNA extraction using the QIAamp DNA Mini Kit (QIAGEN, Duesseldorf, Germany) according to the manufacturer’s instructions.

Each tick was rinsed with PBS, placed into a 2 mL centrifuge tube, and cut using sterilized scissors. Subsequently, 600 μL of PBS buffer (0.1 M, pH 7.4) and two grinding beads were added to the tube. The samples were then homogenized using an ultrasonic grinder at 60 Hz for six cycles of 90 s each. The resulting homogenate was centrifuged at 5500× *g* for 5 min at 4 °C. The supernatant was collected for PCV3 antigen detection and DNA extraction using the Omega E.Z.N.A Viral DNA Kit (Omega Bio-Tek, Norcross, GA, USA) according to the manufacturer’s instructions; the pellet was used for tick 16S rDNA detection, with DNA extracted using the MolPure Cell/Tissue DNA Kit (Yeasen, China) for sequencing and classification.

### 2.3. Conventional PCR for PCV3 Detection in Wild Boar Samples

A conventional PCR was performed to detect PCV3 as previously described [39] within one week of sample collection. A total of 2 µL of extracted DNA was added to a PCR mixture containing 10 µL of 1 × Phire animal tissue PCR buffer, 0.4 µM of each primer (Table 1), 0.4 µL of Phire hot start II DNA polymerase (Invitrogen, Carlsbad, CA, USA), and double-distilled water (ddH_2_O) to a final volume of 20 μL. The PCR cycling conditions were as follows: initial denaturation at 98 °C for 5 min; 45 cycles of denaturation at 98 °C for 5 s, annealing at 68 °C for 7 s, and extension at 72 °C for 15 s, followed by a final extension at 72 °C for 60 s. PCR products were visualized on 1% TAE agarose gel. A recombinant plasmid containing the full genome of the reference PCV3 strain (Accession number: KT869077), synthesized by Genscript (Genscript Biotech Corporation, Nanjing, China), was used as the positive control, while ddH_2_O was used as the negative control.

### 2.4. Detection of PCV3 Antibodies in Wild Boar Sample

A commercial PCV3 ELISA kit (BioStone, Dallas, TX, USA), utilizing plates coated with recombinant PCV3 capsid protein, was used to detect antibodies against PCV3 in serum samples according to the manufacturer’s instructions. Samples were considered positive for PCV3 antibody when the S/P value (S/P = the OD value of the sample divided by the average OD value of the positive control × 100) was ≥0.4.

### 2.5. Morphological and Molecular Identification of Ticks

All collected ticks were identified morphologically using both a stereomicroscope and a light microscope, following established taxonomic keys [41]. All ticks were adults and were initially identified at the genus level. After identification, the samples were stored at −80 °C until DNA extraction. To further confirm species-level classification, representative samples from the identified genera were selected for molecular analysis. The 16S rDNA fragment was amplified using specific primers [41,42]—the forward primer 16SrDNA-F (5′-CTGCTCAATGATTTTTTAAATTGCTGTGG-3′) and the reverse primer 16SrDNA-R (5′-CCGGTCTGAACTCAGATCAAGT-3′). PCR reactions were performed in a 50 µL volume containing 25 µL of 2 × Phanta Max Master Mix (Vazyme Biotech, Shanghai, China), 0.4 µM of each primer, 2 µL of the template DNA, and ddH_2_O to adjust the final volume. The cycling conditions were as follows: initial denaturation at 95 °C for 5 min, followed by 35 cycles of denaturation at 95 °C for 30 s, annealing at 50 °C for 15 s, and extension at 72 °C for 30 s, with a final extension at 72 °C for 5 min. PCR products were analyzed by 1% agarose gel electrophoresis. Amplified products were purified using the E.Z.N.A. Tissue DNA Kit (Omega Bio-tek, Norcross, USA) and sequenced by Tsingke Company (Nanjing, China) via the Sanger method. The obtained 16S rDNA sequences were aligned using the BLASTN tool from NCBI (http://www.ncbi.nlm.nih.gov/BLAST/, accessed on 20 November 2023) against the GenBank database for tick species identification.

### 2.6. RT-qPCR for PCV3 Detection in Ticks

Based on a previous study, RT-qPCR was performed to detect PCV3 genomes in the DNA extracted from ticks [39]. Specific primers and probes (Table 1) were used for amplification. For the reaction system, each 20 µL reaction mixture contained 10 µL of 2 × Premix Ex Taq™ Probe (Takara BIO, Shiga, Japan), 1.2 µL of each primer (10 µM) (Table 1), 0.06 µL of the PCV3-specific probe (100 µM), 2 µL of the template DNA, and ddH_2_O to adjust the final volume. The amplification program was as follows: initial denaturation at 95 °C for 7 min, followed by 45 cycles of denaturation at 96 °C for 10 s and annealing/extension at 60 °C for 30 s. Reactions were conducted on a Quant Studio 3 Real-Time PCR system (Thermo Fisher Scientific, Waltham, MA, USA).

### 2.7. Genome Sequencing and Phylogenetic Analysis

Complete PCV3 genome sequences were obtained using three pairs of primers (Table 1). Each 50 µL PCR reaction contained 1 µL of sample DNA template, 25 µL of Vazyme Lamp Master Mix (Vazyme, Nanjing, China), and 0.4 µM of each primer. The PCR conditions were as follows: pre-denaturation at 94 °C for 5 min, followed by 35 cycles of denaturation at 94 °C for 30 s, annealing for 45 s, and extension at 72 °C for 40 s, with a final extension at 72 °C for 7 min. PCR products were purified using the Gel Extraction Kit (Omega Bio-tek, Norcross, USA) and subsequently cloned into the pEASY-Blunt Simple Cloning Vector (TransGen Biotech, Beijing, China). The recombinant plasmids were submitted to GeneralBiol (Chuzhou, China) for Sanger sequencing. Sequences derived from different PCV3 PCR amplifications were assembled using DNASTAR Lasergene software version 7.1 to obtain the complete target sequence.

The sequences obtained in this study, along with reference strains (Appendix A), were aligned using the ClustalW algorithm. Phylogenetic analysis was conducted with the maximum-likelihood method based on the best-fit Tamura–Nei nucleotide model with Gamma-distributed rates (+G), as implemented in MEGA X 10.1.7 software. Bootstrap values were determined based on 1000 replicates.

### 2.8. Statistical Analysis

The differences in PCV3 detection rates by conventional PCR among different types of wild boar samples were evaluated using Fisher’s exact test, and *p*-values lower than 0.05 were considered statistically significant. All analyses were performed using SPSS Statistics 26.0 software (IBM, Chicago, IL, USA).

## 3. Results

### 3.1. Conventional PCR Results of PCV3 in Wild Boars

A total of 110 samples, including 43 whole blood, 43 serum, 8 fecal, 1 oral fluid, and 15 tissue homogenate samples, were collected from 47 wild boars and tested for PCV3 by PCR. Sixteen wild boars were positive for PCV3, with twelve positives detected in whole blood, one in serum, two in tissue homogenates, and one in fecal samples, resulting in a PCV3-positive rate of 34.0% among the captured wild boars (Table 2). The positive rate in whole blood was significantly higher than that in serum from the same wild boars (*p* < 0.01). PCV3-positive wild boars were distributed across 7 out of the 11 districts in Nanjing (Figure 1).

### 3.2. Antibody Detection Results for PCV3

Serum samples from 43 wild boars were tested for antibodies against PCV3. A total of 18 wild boars were seropositive, resulting in a positive rate of 41.9% (Table 2). The seropositive group (n = 18) showed a mean OD value of 0.596 and a median of 0.572, while the seronegative group (n = 24) exhibited markedly lower values, with a mean of 0.203 and a median of 0.185. When considering all samples, the overall mean and median OD values were 0.371 and 0.311, respectively.

### 3.3. Morphological Identification of Ticks and RT-qPCR Results of PCV3 in Ticks

A total of 83 tick samples were collected from the wild boars. Morphological identification classified the ticks into two genera—*Haemaphysalis* spp. (n = 33) and *Amblyomma* spp. (n = 50) (Appendix A). Based on 16S rDNA sequencing, the ticks were further identified as *Haemaphysalis longicornis*, *Haemaphysalis hystricis*, *Haemaphysalis flava*, and *Amblyomma testudinarium*. Among 83 tick samples analyzed, 6 samples (7.2%) tested positive for PCV3, and all positive ticks were identified as *Amblyomma testudinarium* (Appendix A). Attempts to amplify the PCV3 genome sequences from all positive samples using conventional PCR methods were unsuccessful.

### 3.4. Sequence Analysis of PCV3 in Wild Boar Samples

A complete PCV3 genome sequence (PCV3/CN/Nanjing/WB/2021, accession number: ON359944) was obtained after attempts to amplify the full-length PCV3 genome sequences from 14 PCV3-positive samples. According to the latest genotyping proposal, PCV3 can be classified into two clades: PCV3a and PCV3b [43]. Polygenetic analysis revealed that PCV3/CN/Nanjing/WB/2021 clustered within PCV3a (Figure 2), sharing 98.5% to 99.1% nucleotide identity with PCV3a reference strains. Further genetic analysis showed that PCV3/CN/Nanjing/WB/2021 exhibited 98.9% nucleotide similarity with PCV3 genomes retrieved from wild boars in Brazil (MT075517, MT075518, and MT075519) and 98.9 to 99.1% similarity with PCV3 sequences identified in domestic pigs in China (MK580468, MF069116, and MH491016). In addition, PCV3/CN/Nanjing/WB/2021 shared up to 99.1% genome-wide identity with a PCV3 sequence (MK580468) isolated from domestic pigs in Nanjing (Appendix A).

## 4. Discussion

In the present study, we screened 47 wild boars captured between March 2021 and November 2022 from designated locations in Nanjing, Jiangsu Province, China. PCV3 was detected in multiple specimen types, with PCR results revealing positivity rates of 27.9% (12/43) in whole blood, 4.7% (2/43) in serum, 20.0% (3/15) in tissue homogenates, and 25.0% (2/8) in fecal samples, whereas the single oral fluid sample tested negative (0%, 0/1) (Table 2). The overall detection rate of PCV3 in this wild boar population was 34.0%, which aligns with previous reports from Italy [5], Brazil [44], and Spain [26], where positivity rates ranged from 33.2% to 44.7% and samples were mainly collected from hunted animals or wildlife institutions. Notably, the detection rate observed in our study was significantly higher than the 5.8% previously reported in wild boars of unclear origin in Jiangxi Province, China [45]. This discrepancy may be attributed to differences in sample origin, ecological context, and environmental exposure. In our study, most animals were captured from urban or peri-urban areas, where wild boars are more likely to encounter anthropogenic sources of contamination, including household waste, environmental fomites, and livestock markets. Moreover, the proximity of wild boars to domestic animals and human settlements in urban environments potentially facilitates interspecies interactions, thereby increasing the risk of pathogen spillover and transmission [46,47]. Overall, these findings suggest that the higher population densities and increased interaction among wild boars, domestic animals, and humans in urban areas may contribute to a higher probability of PCV3 infection in wild boars in these areas.

Interestingly, all PCV3 PCR-positive samples were obtained from male wild boars. Only three females were captured and included in the sample set. The predominance of male samples may be attributed to their higher activity levels and greater likelihood of observation compared to females [48]. Given the small sample size of female wild boars, the potential association between sex and PCV3 detection frequency requires further investigation.

In this study, the PCV3-positive rate was significantly higher in whole blood than in serum samples from the same wild boars (*p* < 0.01). One possible explanation for this observation is that PCV3 may replicate or persist in peripheral blood mononuclear cells, which are abundant in whole blood [49], thereby leading to higher concentrations of viral DNA during extraction due to the larger amount of cellular material present [50].

In this study, a complete PCV3 genome sequence was successfully recovered, specifically from a tissue homogenate sample. Consistent with a previous report, tissue samples tend to harbor higher PCV3 viral loads compared to serum [26], suggesting that selecting samples with higher viral loads improves the likelihood of successful genome recovery. Notably, the PCV3/CN/Nanjing/WB/2021 strain shared up to 99.1% genome-wide identity with a PCV3 strain (MK580468), which is the most similar sequence available in Jiangsu Province and was previously identified in domestic pigs from Nanjing, suggesting potential viral transmission between wild boars and domestic pigs within the region. These findings imply an epidemiological connection between wild and domestic swine populations and underscore the importance of further investigations into the transmission dynamics of PCV3 in China. Furthermore, the selection of a sequencing method can significantly influence genetic analysis. In the present study, conventional PCR was used for segmental amplification of the complete genome. However, due to its requirement for high viral load, the amplification efficiency was relatively low. In contrast, other studies have successfully obtained the PCV3 genome sequences through metagenomic sequencing methods [1,2], which bypass the limitations of amplification and are less dependent on viral load.

Globally, serological surveillance data for PCV3 in wild boars remain limited, primarily due to the challenges associated with collecting serum samples from live animals. In this study, IgG antibodies against the PCV3 capsid protein were detected in 18 of 43 (41.9%) serum samples (Table 2), a rate higher than the 18.3% positive rate reported in a recent study involving 60 wild boars from China. That study also utilized an indirect ELISA based on the PCV3 capsid protein for antibody detection [51]. However, detailed information regarding the sample collection date, locations, or concurrent PCV3 genome detection was not provided in the previous report. The serological and genomic surveillance results from the current study collectively confirm the high level of PCV3 exposure among wild boars in Nanjing.

In this study, 83 ticks collected from captured wild boars were screened for PCV3 using qPCR. Six of them tested positive, all identified as *Amblyomma testudinarium* (family Ixodidae, genus *Amblyomma*). *Amblyomma testudinarium* is widely distributed throughout East and Southeast Asia, particularly in warm and humid environments [52]. This species parasitizes a wide range of mammalian hosts, including wild boars, brown bears, and domestic dogs [53,54,55]. It has been confirmed to harbor various pathogens, such as the Severe Fever with Thrombocytopenia Syndrome virus (SFTSV), *Rickettsia* spp., and *Borrelia burgdorferi* [56,57,58]. Due to the low viral loads (Appendix A), PCV3 gene amplification from the positive tick samples was unsuccessful, precluding direct sequence comparison between the PCV3 strains identified in wild boars and ticks. In this study, PCV3-positive ticks were collected from three wild boars, of which wild boar No. 8 tested positive for DNA but negative for antibodies, and wild boar No. 9 tested positive for both DNA and antibodies. Interestingly, wild boar No. 45 tested negative for PCV3 DNA but positive for antibodies. Similarly, a prior study identified PCV3 in two ticks (*Ixodes ricinus*) collected from PCV3-negative roe deer [35]. Several hypotheses could explain these findings: (1) the tick may have previously fed on a PCV3-infected host, detached, and later parasitized the sampled wild boar; (2) PCV3 may persist within the tick for extended periods, allowing detection even after viral clearance from the host’s bloodstream; or (3) the detection of PCV3 in the tick may represent a false positive, as the possibility cannot be completely ruled out despite repeated testing. Although conclusive evidence is still lacking regarding the role of ticks as vectors for PCV3 transmission, the findings of this study demonstrate that PCV3 can be detected within ticks, suggesting a potential role in the viral dissemination. To better understand the possible role of ticks in PCV3 transmission, further studies are needed to assess PCV3 survival, replication, and transmission efficiency in ticks.

One limitation of this study is the use of different detection methods for wild boars and ticks. Given the extended duration of this study, conventional PCR—previously employed for PCV3 detection in wild boars [35]—was consistently used for wild boar samples to ensure comparability with earlier reports. However, when ticks became available for sampling, the limited amount of extractable viral nucleic acid and the necessity of concurrent 16S rDNA identification prompted the adoption of a more sensitive qPCR assay validated in our laboratory for PCV3 detection. This methodological discrepancy may introduce variability in detection rates and should be considered when interpreting the results.

In conclusion, this study reports the detection of PCV3 in wild boars and *Amblyomma testudinarium* ticks parasitizing wild boars in Jiangsu Province, China. These findings suggest that wild boars may serve as potential reservoirs and ticks as potential vectors for PCV3, highlighting the need for further research to better understand their roles in the transmission and maintenance of swine pathogens.

## Figures and Tables

**Figure 1 viruses-17-01049-f001:**
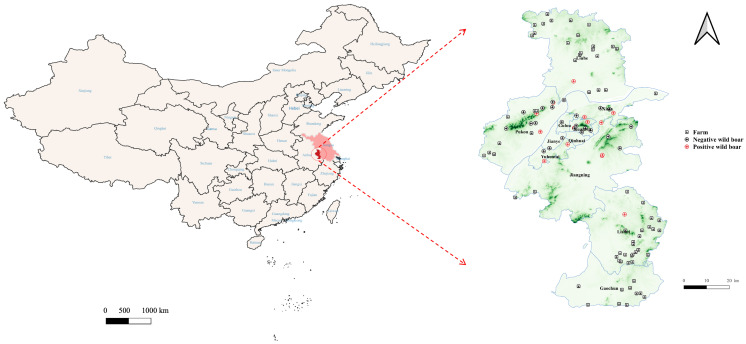
Distribution of captured wild boars and swine farms (≥50 heads) in Nanjing, Jiangsu Province. Two PCV3-positive and two PCV3-negative wild boars lacked coordinate data. The map was constructed with QGIS software version 3.20 (QGIS, 2020). The dark red area represents Nanjing City, and the light red area indicates Jiangsu Province.

**Figure 2 viruses-17-01049-f002:**
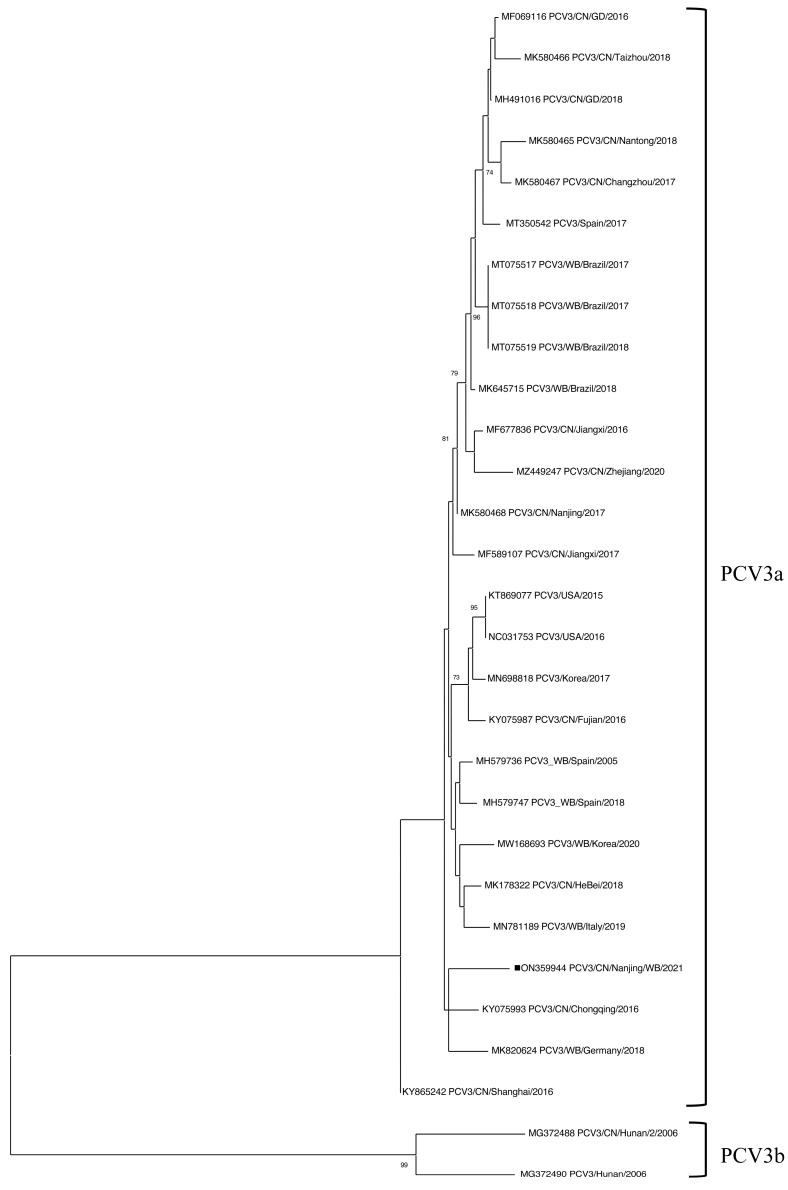
Phylogenetic trees based on the complete genome of PCV3 and constructed by the maximum-likelihood method in MEGA X software. Bootstrap values were based on 1000 replicates and showed each node if >70%. The PCV3/CN/Nanjing/WB/2021 strain identified in this study is highlighted with a black rectangle.

**Table 1 viruses-17-01049-t001:** Primers and probe used for the PCV3 detection and complete genome sequencing in the conventional PCR and RT-qPCR for wild boars and ticks.

Primer	Sequence (5′–3′)	PCR Product	Assay	Reference
PCV3-74F	CACCGTGTGAGTGGATATAC	1072 bp	Conventional PCR	[40]
PCV3-1144R	CACCCCAACGCAATAATTGTA			
PCV3-1137F	TTGGGGTGGGGGTATTTATT	425 bp		
PCV3-1561R	ACACAGCCGTTACTTCAC			
PCV3-1427F	AGTGCTCCCCATTGAACG	1007 bp		
PCV3-433R	CGACCAAATCCGGGTAAGC			
PCV3-qF	TGACGGAGACGTCGGGAAAT	113 bp	RT-qPCR	[39]
PCV3-qR	CGGTTTACCCAACCCCATCA			
PCV3-probe	FAM-GGGCGGGGTTTGCGTGATTT-BHQ1			

**Table 2 viruses-17-01049-t002:** Sample collection data and PCV3 detection results for each wild boar.

Wild Boar ID	PCV3 PCR Results						PCV3 Antibody Results
	Blood	Serum	Feces	Tissues	Oral Fluid	Ticks	
1	NC *	NC	−	NC	−	NC	NC
2	+ **	−	+	NC	NC	NC	+
3	−	−	−	NC	NC	NC	+
4	+	−	−	NC	NC	NC	−
5	−	−	−	NC	NC	0/5	+
6	+	−	NC	−	NC	0/11	−
7	+	−	NC	−	NC	0/3	−
8	+	+	NC	+	NC	2/2	−
9	+	−	−	NC	NC	3/18	+
10	+	−	NC	NC	NC	NC	+
11	NC	NC	+	NC	NC	NC	−
12	+	−	−	NC	NC	NC	+
13	−	−	NC	−	NC	NC	+
14	−	−	NC	−	NC	NC	+
15	+	−	NC	NC	NC	NC	+
16	−	−	NC	NC	NC	NC	+
17	−	−	NC	NC	NC	NC	−
18	−	−	NC	−	NC	NC	+
19	+	−	NC	NC	NC	0/4	+
20	+	−	NC	−	NC	NC	−
21	−	+	NC	−	NC	NC	+
22	+	−	NC	NC	NC	0/7	−
23	−	−	NC	NC	NC	NC	+
24	−	−	NC	NC	NC	NC	+
25	−	−	NC	−	NC	NC	−
26	−	−	NC	NC	NC	NC	−
27	−	−	NC	NC	NC	NC	−
28	−	−	NC	NC	NC	NC	+
29	−	−	NC	NC	NC	NC	−
30	−	−	NC	NC	NC	NC	−
31	−	−	NC	NC	NC	NC	−
32	−	−	NC	−	NC	NC	−
33	−	−	NC	−	NC	NC	−
34	−	−	NC	−	NC	0/6	+
35	−	−	NC	NC	NC	NC	−
36	−	−	NC	NC	NC	NC	−
37	−	−	NC	NC	NC	NC	−
38	−	−	NC	NC	NC	NC	−
39	−	−	NC	NC	NC	NC	−
40	−	−	NC	NC	NC	NC	−
41	−	−	NC	+	NC	NC	−
42	−	−	NC	NC	NC	NC	−
43	−	−	NC	NC	NC	NC	−
44	−	−	NC	NC	NC	NC	−
45	−	−	NC	NC	NC	1/11	+
46	NC	NC	NC	−	NC	NC	NC
47	NC	NC	NC	+	NC	0/16	NC
Total	12/43	2/43	2/8	3/15	0/1	6/83	18/43

* NC: not collected; ** “−” indicates PCV3-negative wild boars; “+” indicates PCV3-positive wild boars.

## Data Availability

The original contributions presented in this study are included in the article/Appendix A. Further inquiries can be directed to the corresponding author.

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
