# Peer review of "Detection of Porcine Circovirus Type 3 in Free-Ranging Wild Boars and Ticks in Jiangsu Province, China"

_viruses, 2025, doi:10.3390/v17081049_

Round 1
Reviewer 1 Report (Previous Reviewer 1)
Comments and Suggestions for Authors
The manuscript is improved, and my comments are properly addressed.
Author Response
Comments 1: The manuscript is improved, and my comments are properly addressed.
Response 1: We sincerely thank for the positive feedback on the revised manuscript and for acknowledging that the previous comments have been addressed.
Reviewer 2 Report (Previous Reviewer 3)
Comments and Suggestions for Authors
Thanks for the responses. I still have concerns to be addressed.
- Due to the poor quality of the figures, I was unable to verify whether the results description is accurate. The current figures remain unacceptable for publication. Please replace them with higher-resolution versions.
- In Lines 266–267, “clade 1” should be corrected to “PCV3a” for consistency with the classification used in the study.
- Suggest providing a clear version of this manuscript for peer-review.
Author Response
Comments 1: Due to the poor quality of the figures, I was unable to verify whether the results description is accurate. The current figures remain unacceptable for publication. Please replace them with higher-resolution versions.
Response 1: We sincerely apologize for the figure resolution in the original submission. Thank you for highlighting this critical issue. We have now replaced figures with high-resolution versions.
Comments 2: In Lines 266–267, “clade 1” should be corrected to “PCV3a” for consistency with the classification used in the study.
Response 2: We thank the reviewer for highlighting this inconsistency. The term "clade 1" has been corrected to "PCV3a" to align with the classification system consistently used throughout the manuscript.
Comments 3: Suggest providing a clear version of this manuscript for peer-review.
Response 3: Thank you for this suggestion. We have now uploaded a revised manuscript version.
Reviewer 3 Report (New Reviewer)
Comments and Suggestions for Authors
This study assesses the presence of PCV3 in wild boar and ticks in Jiangsu, China. Because PCV3 is a virus of relatively recent discovery and of importance for swine production, this study provides useful information on the distribution of the virus. The manuscript is well written and does not have major flaws. I would recommend the following minor changes:
- Line 41: should be clinical signs instead of clinical symptoms.
- Line 135: why were ticks dissected if they were then ground? Maybe the authors meant that the ticks were cut in half?
- Table 1: please check for errors. There are only 3 primers labeled F and 5 labeled R.
- Results: please report how many samples were attempted for sequencing.
- Lines 358-360: The first and second factors appear to be the same.
- I propose to remove lines 361-367 as they do not add much to the discussion.
- MK580468 appears to be the closest sequence in GenBank, however this is never specifically reported. Please clarify this in the discussion. If that is not the case, please report the closest match in a publicly available database.
- Lines 410-412: replace antigen with DNA
Author Response
Comments 1: Line 41: should be clinical signs instead of clinical symptoms.
Response 1: We thank the reviewer for this suggestion. The phrase "clinical symptoms" in Line 41 has been revised to "clinical signs".
Comments 2: Line 135: why were ticks dissected if they were then ground? Maybe the authors meant that the ticks were cut in half?
Response 2: Thank you very much for your valuable question. The term "dissected" mentioned in the text actually refers to appropriately cutting the ticks. The purpose of this operation is to destroy the tick's body wall structure, allowing the beads to make more sufficient contact with the tick's tissues during the subsequent grinding process. We realize that the original expression may be ambiguous. We have revised it to "Each tick was rinsed with PBS, placed into a 2 mL centrifuge tube and cut using sterilized scissors" to more accurately reflect the operation process. Thank you for your careful review.
Comments 3: Table 1: please check for errors. There are only 3 primers labeled F and 5 labeled R.
Response 3: We sincerely thank the reviewer for detecting this labeling inconsistency in Table 1. The corrections have been implemented.
Comments 4: Results: please report how many samples were attempted for sequencing.
Response 4: Thank you for your comment. We attempted to amplify the relevant sequences in 6 tick samples and 14 wild boar samples. We have added this information to Results 3.3 and 3.4.
Comments 5: Lines 358-360: The first and second factors appear to be the same.
Response 5: Thank the reviewer for this suggestion. The original two factors have been consolidated into a unified explanation to eliminate repetition in Line 256-259 with highlight.
Comments 6: I propose to remove lines 361-367 as they do not add much to the discussion.
Response 6: We appreciate the reviewer's suggestion. The content in Lines 361-367 has been removed as proposed.
Comments 7: MK580468 appears to be the closest sequence in GenBank, however this is never specifically reported. Please clarify this in the discussion. If that is not the case, please report the closest match in a publicly available database.
Response 7: Thank you for your insightful comment. We have revised the discussion section to clarify that MK580468 is indeed the most similar sequence available in GenBank from Jiangsu Province in Line 264-267. This revision has been made to highlight the potential epidemiological link between wild boars and domestic pigs in the region.
Comments 8: Lines 410-412: replace antigen with DNA
Response 8: We thank the reviewer for this terminology correction. The term "antigen" in Lines 410-412 has been replaced with "DNA" .
Reviewer 4 Report (New Reviewer)
Comments and Suggestions for Authors
This study investigated the presence and genetic characteristics of Porcine Circovirus Type 3 (PCV3) in wild boars and parasitizing ticks in Nanjing, Jiangsu Province, China. From March 2021 to November 2022, samples including blood and serum from 47 wild boars, as well as 83 ticks, were tested. The results showed that 34.0% (16/47) of the wild boars tested positive for PCV3 by PCR, and 41.9% (18/43) of the wild boars were seropositive for antibodies. Additionally, 7.2% (6/83) of the ticks tested positive by RT-qPCR, all of which were identified as Amblyomma testudinarium. The PCV3 genome obtained from wild boars was classified as PCV3a, showing a high homology (99.1%) with the strain from domestic pigs in Nanjing, suggesting a possible transmission relationship of PCV3 among wild boars, domestic pigs and ticks. But,the clarity of the figures in the article is relatively low, and it is recommended to revise them before publication.
Author Response
Comments 1: This study investigated the presence and genetic characteristics of Porcine Circovirus Type 3 (PCV3) in wild boars and parasitizing ticks in Nanjing, Jiangsu Province, China. From March 2021 to November 2022, samples including blood and serum from 47 wild boars, as well as 83 ticks, were tested. The results showed that 34.0% (16/47) of the wild boars tested positive for PCV3 by PCR, and 41.9% (18/43) of the wild boars were seropositive for antibodies. Additionally, 7.2% (6/83) of the ticks tested positive by RT-qPCR, all of which were identified as Amblyomma testudinarium. The PCV3 genome obtained from wild boars was classified as PCV3a, showing a high homology (99.1%) with the strain from domestic pigs in Nanjing, suggesting a possible transmission relationship of PCV3 among wild boars, domestic pigs and ticks. But,the clarity of the figures in the article is relatively low, and it is recommended to revise them before publication.
Response 1: We sincerely thank the reviewer for the feedback on our study. We have replaced figures with high-resolution versions.
Round 2
Reviewer 2 Report (Previous Reviewer 3)
Comments and Suggestions for Authors
I have carefully reviewed the files you provided. The figures are of sufficient quality for publication. Based on my evaluation of all submitted materials, I recommend acceptance in their current form.
This manuscript is a resubmission of an earlier submission. The following is a list of the peer review reports and author responses from that submission.
Round 1
Reviewer 1 Report
Comments and Suggestions for Authors
A collection of 110 different types of samples from 49 wild boars were tested for the presence of PCV3. A full genome of PCV3 was sequenced from a positive sample and compared with those from domestic pigs and showed very high homology indicating cross transmission between wild and domestic animals. Although low PCV3 positive rate was observed from the ticks collected from these animals, the fact that PCV3 was identified from the ticks provided the evidence that PCV3 can possibility be transmitted by ticks.
The manuscript is generally well-written. I am providing some additional suggestions as listed below.
Replace all “;” with “,” in the abstract
Line 25 (L25): homology between strains from the wild and domestic pigs.
L31: clinically identified ….pigs, are PCV1,…
L55-56: common vectors transmitting…
L58: found to be positive..
Comments on the Quality of English LanguageThe quality of the manuscript is generally good, and well-written with some minor improvements.
Reviewer 2 Report
Comments and Suggestions for Authors
The submitted manuscript is a report of the detection of PCV3 in wild boars and ticks in Jiangsu Province, China. The manuscript describes the use of PCR and serology to identify the infection and seroconversion of PCV3 in the wild boar and tick populations. Significant weaknesses lie in the fairly limited data presented and that two different PCRs were used for the detection of PCV3. One for the boars and one for the ticks. The provided rationale was that one was more sensitive than the other, then why not use the more sensitive PCR for all sample types? Strengths lie in the number of animals tested and the novelty of finding PCV3 in ticks. Genetic sequencing of PCV3 was performed and compared with one sample. This manuscript would be better off submitted as a brief communication.
Line 30 – 32: Please reword this sentence for clarity.
Section 2.1: Since some boars were catch and release, is it possible that the same boar was recaptured and thus skewed results?
Line 124 – 126: I believe that the acronym should be S/P not P/P as in sample OD/positive control OD.
Section 3.2: What were the mean and median OD values for the PCV3 serology?
Line 295 -297: When was PCV3 antigen tested for? This is not described in the methods.
Comments on the Quality of English LanguageThe quality of the written English is fair with only a few instances of rewriting needed.
Reviewer 3 Report
Comments and Suggestions for Authors
The manuscript by Sun et al. investigated the presence and genetic characteristics of Porcine Circovirus Type 3 (PCV3) in free-ranging wild boars and ticks in Jiangsu Province, China. This study proposes a relationship between PCV3 infection in domestic pigs, wild boars, and ticks, emphasizing potential transmission pathways at the wildlife-livestock-human interface. Although PCV3 was discovered in ticks, the low viral load and lack of effective genome sequencing from tick samples provide little evidence that ticks act as transmission vectors. Additional research is required to demonstrate a solid link.
Specific comments:
- What are your standards of reference sequence selection? Why only one clade 2 sequence was selected? Having only one reference sequence for clade 2 in the phylogenetic analysis weakens the conclusions regarding the phylogenetic placement of the study's PCV3 sequence and the overall evolutionary relationships within the PCV3 clades.
- As shown in lines 213-215 “PCV3/CN/Nanjing/WB/2021 displayed a genome-wide similarity of up to 99.1% with a PCV3 sequence (MK580468) identified in domestic pigs in Nanjing”, but phylogenetic tree analysis shown a closer relationship with MK820624 from Germany. Why? Please discuss more.
- Figures qualities are unacceptable, please replace theses by higher resolution ones.
- Please add the morphological identification results of ticks.
The English should be improved to more clearly express the research. Please use the English Editing service to polish this manuscript.